# Release of Interferon-β (IFN-β) from Probiotic *Limosilactobacillus reuteri*-IFN-β (LR-IFN-β) Mitigates Gastrointestinal Acute Radiation Syndrome (GI-ARS) following Whole Abdominal Irradiation

**DOI:** 10.3390/cancers15061670

**Published:** 2023-03-08

**Authors:** Diala F. Hamade, Michael W. Epperly, Renee Fisher, Wen Hou, Donna Shields, Jan-Peter van Pijkeren, Amitava Mukherjee, Jian Yu, Brian J. Leibowitz, Anda M. Vlad, Lan Coffman, Hong Wang, M. Saiful Huq, Ziyu Huang, Claude J. Rogers, Joel S. Greenberger

**Affiliations:** 1Department of Radiation Oncology, UPMC Hillman Cancer Center, Pittsburgh, PA 15232, USA; 2Department of Food Science, University of Wisconsin-Madison, Madison, WI 53706, USA; 3Department of Pathology, University of Pittsburgh, Pittsburgh, PA 15260, USA; 4Department of OB/Gyn and Reproductive Sciences, University of Pittsburgh, Pittsburgh, PA 15260, USA; 5Department of Medicine, University of Pittsburgh, PA 15260, USA; 6Department of Biostatistics, University of Pittsburgh, Pittsburgh, PA 15260, USA; 7ChromoLogic, LLC, Monrovia, CA 91016, USA

**Keywords:** *Lactobacillus reuteri*, irradiation, Interferon-β (IFN-β)

## Abstract

**Simple Summary:**

The aim of our study was to assess the efficacy of a novel approach that combines whole abdomen irradiation (WAI) with genetically engineered *Limosilactobacillus reuteri*, known as *Lactobacillus reuteri* until recently, releasing interferon-beta (IFN-β; LR-IFN-β). We established that LR-IFN-β is a potent intestinal radioprotector and mitigator, and engineered bacteria are rapidly cleared from the digestive tract following WAI. The impact of our proposal is to improve the survival of ovarian cancer (OC) patients by adding WAI combined with other treatments as a possible treatment. We plan to further increase the survival and quality of life for OC patients.

**Abstract:**

Irradiation can be an effective treatment for ovarian cancer, but its use is limited by intestinal toxicity. Thus, strategies to mitigate toxicity are important and can revitalize the current standard of care. We previously established that LR-IL-22 protects the intestine from WAI. We now hypothesize that LR-IFN-β is an effective radiation protector and mitigator and is rapidly cleared from the digestive tract, making it an option for intestinal radioprotection. We report that the gavage of LR-IFN-β during WAI provides improved intestinal barrier integrity and significantly preserves the numbers of Lgr5+GFP+ intestinal stem cells, improving survival. The rapid clearance of the genetically engineered probiotic from the digestive tract renders it a safe and feasible radiation mitigator. Therefore, the above genetically engineered probiotic is both a feasible and effective radiation mitigator that could potentially revolutionize the management of OC patients. Furthermore, the subsequent addition of platinum/taxane-based chemotherapy to the combination of WAI and LR-IFN-β should reduce tumor volume while protecting the intestine and should improve the overall survival in OC patients.

## 1. Introduction

Previous interest in whole abdomen irradiation (WAI) as an adjuvant therapeutic modality for the treatment of abdominal tumors, such as ovarian cancer, has decreased due to significant acute intestinal toxicity [1,2,3,4,5,6,7,8,9,10], as well as injuries to the hepatic, renal, and abdominal lymphoid organs. More recent clinical trials of WAI using a low-dose fraction size have yielded suboptimal results [8,9]. Thus, intestinal radioprotectors are needed that do not compromise therapeutic tumor cytoreduction [11,12,13,14,15,16,17]. In a mouse model of WAI in ovarian cancer, we reported that the oral administration (gavage) of a second-generation probiotic *L. reuteri* (LR) that was engineered to synthesize interleukin-22 (LR-IL-22) is a safe and effective intestinal radioprotector [18,19,20]. LR-IL-22 gavage improved intestinal barrier integrity following fractionated WAI and further improved the survival of 2F8cis tumor-bearing mice. Furthermore, LR-IL-22 enhanced the therapeutic effectiveness of first-line chemotherapy [18]. We recently reported that IFN-β treatment via intraperitoneal (IP) injection 48 h after irradiation improves survival and enhances Lgr5+ stem cell numbers and the regeneration of intestinal crypts following total body irradiation (TBI) [21]. In the present studies, we tested whether genetically engineered *L. reuteri* releasing IFN-β (LR-IFN-β) is an effective radiation protector and mitigator for intestinal crypt stem cells and if this therapeutic drug increases mouse survival after exposure to irradiation doses that cause gastrointestinal acute radiation syndrome (GI-ARS). Analogous to what we previously observed with LR-IL-22, here, we report that following the oral administration of LR-IFN-β 24 h after irradiation, LR-IFN-β is cleared from the intestine within 3 days following irradiation while also being a potent mitigator of GI-ARS. Our results also show that LR-IFN-β is an effective radiation mitigator and may be as effective as LR-IL-22. These results establish that LR-IFN-β is a potentially valuable intestinal radiation mitigator and radioprotector and may facilitate the addition of therapeutic WAI to new protocols for the treatment of ovarian cancer.

## 2. Materials and Methods

### 2.1. Mouse Model

Adult male and female C57BL/6NTac mice (Taconic Biosciences, Inc., Germantown, NY, USA) and C57BL/6 mice with knock-in allele-Lgr5-EGFP-IRES creERTZ [22] (Lgr5+GFP+) were housed 4 mice per cage and fed standard laboratory chow and deionized drinking water. All mice were treated according to University of Pittsburgh Institutional Animal Care and Use Committee regulations (IACUC)-approved protocol. Veterinary care was provided by the Division of Laboratory Resources of the University of Pittsburgh.

### 2.2. Mouse Irradiation

Adult mice received a single fraction of either 9.25 Gy or 12 Gy total body irradiation (TBI) or 13.25 Gy (male) or 13.4 Gy (female) partial-body irradiation (PBI) or 15 Gy whole abdomen irradiation (WAI). PBI and WAI was delivered using a Varian TrueBeam Linear Accelerator (Palo Alto, CA). WAI was administered via a 3 × 30 cm field, with the mice anesthetized with nemubtal and placed so that only the abdomen was in the irradiation field. PBI was administered with the right leg removed from the field. WAI and PBI were given at a source-to-surface distance (SSD) of 100 cm, with a dose rate of 600 monitor units per minute using 6 MV photons. TBI was performed by placing 5 mice in a radiation pie plate and irradiating at 300 cGy per minute in a JL Shepherd Model 68 cesium irradiator (JL Shepherd, Inc., San Fernando, CA). Radiation dosimetry and field accuracy and reproducibility were performed using previously published methods [23]. All radiation parameters followed previously published protocols [23,24].

### 2.3. Construction of LRΔthyA:rpoB(H488R)/pIFNβ-thyA

To construct the LR-IFN-β, we used single-stranded DNA recombineering to inactivate thyA in *L.reuteri* VL1014 [LR:*rpoB*(H488R)]. This bacterium is a rifampicin-resistant derivative to yield a LRΔ*thyA* (Rif^®^), as previously described [25,26]. We replaced the chloramphenicol resistance gene in the pVPL31126 with a thyA gene isolated from *L. reuteri* VPL1014 using blunt-end ligation (T4 DNA ligase, Thermo Fisher Scientific, Waltham, MA, USA). Electroporation was used to transform the LR* into LR*/IFNβ-thyA as previously described, except we substituted IFN-β for IL-22 (26). As an empty vector control, we used a previously constructed vector pCtl-thyA placed in LR* [26]. For an intermediate cloning host, we used E. coli EC1000.

### 2.4. Oral Gavage of LR-IFN-β, Control LR, or Intraperitoneal Delivery of IFN-β Protein

A total of 100 µL suspensions of LR-IFN-β or LR were prepared to contain 10^6^ to 10^11^ colony-forming units of either LR-IFN-β or LR, with an empty plasmid with no IFNβ gene; these were measured by optical density and were gavaged in 100 µL saline. Other mice received IP injections of 25 µg/kg IFN-β at 24 h after irradiation (Pepro-Tech, Rocky Hill, NJ, USA, E. coli, 210–22). Mice were randomly divided into groups: control mice or mice treated with intraoral gavage of either 10^9^ control empty vector L. reuteri (LR) or LR-IL-β cells in 200 µL of saline or IP injections of 20 µg/mL of IL-IFN-β in 100 µL of saline.

### 2.5. Protein Analysis in Plasma and Small Intestine (Ileum) by Luminex Assay

The methods for the preparation of intestinal tissues and plasma for Luminex assay and the production of Luminex chips for 33 inflammatory responses to stress-associated proteins were utilized as published [23,24] (Table 1). The analyses of the data and statistical evaluations were performed as previously published [23]. Briefly, the Luminex Protein Assay plasma was thawed, vortexed, and centrifuged to remove particulate matter prior to dilution. Ileum was removed, and 4 mg was homogenized in 1 mL of phosphate-buffered saline (PBS) containing 0.1% Tween 80 to prevent protein clumping, and stored at −80 °C. Before using in the Luminex Protein Assay, the homogenate was thawed to room temperature and centrifuged at 2000 rpm at 4°C for 10 min. A Bio-Rad protein assay (Bio-Rad Laboratories, Hercules, CA, USA) was used to quantitate the amount of protein in the sample.

### 2.6. Luminex Assay Kits

To measure TGF-β1 expression, we utilized a TGF-β1 Single Plex Magnetic Bead Kit (TGFBMAG-64K-01, EMD Millipore, Billerica, MA, USA). A 32 Multiplex Mouse Cytokine/Chemokine Magnetic Bead Panel (MCYTOMAG-70K Millipore, Billerica, MA, USA) was used to measure the expression of 32 cytokine/chemokine, as shown in Table 1.

### 2.7. Preparation of Reagents and Standards for Immunoassay Bead for Luminex Assays: Quality Controls, Wash Buffer, and Serum Matrix

To prepare for the Luminex assays, antibody-immobilization beads were sonicated for 30 s and then vortexed for 1 min. A total of 250 microliters of assay buffer was used to reconstitute Quality Control 1 and 2. A 1:10 dilution of 60 mL of wash butter in 540 mL of deionized water was performed. For the 32 Multiplex Assay, the plasma samples were diluted into 2.0 mL of Assay Buffer and incubated at room temperature for 10 min.

For the 32 Multiplex Assay, the mouse cytokine standards were reconstituted in 250 microliters deionized water, inverted to mix, vortexed, and then transferred to a polypropylene microfuge tube. Serial dilutions were performed using assay buffer, diluting the standards serially by adding 50 microliters of the standards to 200 microliters of assay buffer, as described in the kit. Similar dilutions were made for the TGFβ1 standard, except the original standard, which was diluted in 150 microliters of assay buffer.

### 2.8. Procedures for the 32-Multiplex Luminex Immunoassay

Subgroups of male and female C57BL/6 mice were sacrificed at serial times postirradiation using carbon dioxide inhalation followed by cervical dislocation according to an IACUC protocol. Blood was obtained by submandibular bleeding in heparinized tubes before sacrifice. Immediately after sacrifice, the ileums were excised immediately after sacrifice, flash frozen in liquid nitrogen, and processed using the Luminex assay, as described above. Specimens were obtained, beginning on day 0, but prior to irradiation (day 0) and daily for the next 5 days, the specimens were collected for a total of 6 time points. Intestines were washed free of blood by injecting 10 mL of PBS into the right ventricle of the heart prior to obtaining the ileum. The Luminex assay for the plasma and intestine was performed in 96-well plates. The 96-well plate was conditioned by first adding 200 microliters of wash buffer to each well of the 96-well plate. The plate was then sealed and covered on a plate shaker for 10 min at room temperature. Wash buffer was then removed via suction.

As in previous publications [18,23,24] the protein concentrations for plasma were presented as pg/mL. For the intestines, the data from the Luminex assay was presented as pg/mg protein. A comparison of samples is reported as upregulated (red color) or downregulated (green color) based on P values of significant differences.

### 2.9. Measurement of Intestine Levels of Tight Junction Proteins I-CAM and Occludin

C57Bl/6NTac mice were divided into 5 groups [0 Gy; 12 Gy TBI; 12 Gy TBI + LR; 12 Gy TBI + IFN-β protein; and 12 Gy TBI + LR-IFN-β]. IFN-β1 protein or IL-22 protein was administered at 24 h post-irradiation. On day 5 post-irradiation, the mice were euthanized and the ileum excised, fecal contents removed by injection with 1 mL of PBS, fixed in 10% formalin, and sectioned. All antibody mixtures were prepared in 5% bovine serum albumin as suggested by the manufacturer. Tissue sections of the ileum were incubated with primary antibody mixture overnight at 4°C then washed with PBS three times for 5 min each. Secondary antibody solutions were added to the sections for 1 h at room temperature. The sections were washed three times with PBS for 5 min each, 0.5 µg/mL DAPI in PBS was added for 10–20 min at room temperature to label nuclei. Fluorescence microscopes at the University of Pittsburgh Center for Biologic Imaging were used to photograph 30 sections from each mouse, which were then subjected to analysis. For each data point, 5 mice were analyzed. Data were presented as mean± standard error of the percent of cells positive for target protein. The primary antibodies were anti-occludin rabbit antibody (cat.no NBP1-87 402, Novus Biologicals, Littleton, CO, USA) and anti-ICAM mouse antibody (Code No. 270415, Seikagaku Corporation, Tokyo, Japan). The secondary antibodies were anti-mouse or anti-rabbit IgG Alexa 488 or 594 (Invitrogen/Fisher Scientific, Waltham, MA, USA).

### 2.10. Assay for Intestinal Lgr5+ Stem Cells

The assay was performed using C57BL/6 mice with the knock-in allele lgr5-EGFP-IRES creERTZ [22], which allows for the counting of green Lgr5+ stem cells in the intestine [22]. Lgr5+ cells were counted by using the green color and using antibodies and assays according to published methods [21,22,27,28].

### 2.11. Assay for Intestinal Crypt Regeneration

BrdU labeling was carried out, as previously published [21], by administering BrdU 2 h before irradiation.

### 2.12. Dose-Response Curve of Gavaged LR-IFN-β, Measuring Survival after 13.5 Gy Partial-Body Irradiation

In 200 µL saline, different numbers of LR or LR-IFN-β bacterial cells, ranging from 10^6^ to 10^11^ LR-IFN-β, were gavaged into groups of 15 mice per group, as published previously [18,20,21].

### 2.13. Measurement of Clearance of Gavaged LR-IFN-β in Male and Female Mice after 13.25 or 13.4 Gy PBI, Respectively

Fecal samples, intestinal tissue (ileum), spleen, liver, kidney and plasma were collected daily after the gavage of bacteria for 5 days and assayed for the presence of IFN-β protein by Elisa and bacterial colony numbers. Blood was collected by submandibular bleeding. The mice were sacrificed and perfused by injecting 10 mL of PBS through the right ventricle. The intestine was removed and flushed with 1 mL of PBS to remove fecal material. The intestine, fecal material, liver, spleen, kidney, and plasma were flash-frozen in liquid nitrogen. The tissues were thawed and homogenized in 1 mL PBS. The concentration of IFN-β protein was determined by IFN-β ELISA (ab119557, Abcam, Cambridge, MA, USA). For the colony assay the homogenates were diluted and the dilutions were plated on agarose plates containing 20 mM erythromycin. The LR-IFN-β has an erythromycin-resistant gene, so the cells can form colonies. The plates were incubated at 37 °C in the absence of oxygen, and the number of colonies were counted 48 h after plating.

Timeline for experimental protocols. A timeline for the irradiation and tissue collection is shown in Figure 1.

### 2.14. Physics

Linear accelerator-based TBI, PBI (one-leg shielded), and WAI were performed as previously described in detail (19) using either a JL Shepherd Cesium irradiator or a TrueBeam (Varian) linear accelerator with beam flatness and the dose rate monitored daily. For PBI, the dose under the shielded limb was confirmed by thermoluminescent dosimeters (TLDs) to be <1.0% of the dose prescribed to the abdominal cavity. Dose rate was 300 cGy per minute for the cesium irradiator and 600 monitor units at an SSD of 100 cm for the Varian TrueBeam Irradiator. Beam flatness was measured daily. The mice were irradiated with 6 MV photons at a dose rate of 600 MU/min at 100 cm SSD. A strict quality assurance and quality control program was in place to ensure that all beam parameters conform to the guidelines set by the American Association of Physicists in Medicine.

### 2.15. Statistics

For the cytokine data analyses, first, we separately analyzed each radiation treatment (TBI, PBI, or WAI) for each of the 33 cytokines and for each tissue (plasma or intestine). In these analyses, data were summarized for each subgroup using the mean ± standard deviation (SD). On each day of measurement, we compared each treated group with the control. Also, for each treatment, we compared each day with day 0. All these comparisons were carried out based on a two-way ANOVA model, followed by a *t*-test using the estimate statement in Proc GLM of SAS 9.4 (SAS Institute, Inc., Cary, NC, USA). In this ANOVA model, treatment, day, and their interaction were the factors.

For the intestinal crypt stem cell Lgr5+ cell analysis, we also summarized the data using mean ± SD and analyzed the data using two-way ANOVA followed by *t*-tests for the comparisons. For mouse survival, data collection was performed using Kaplan–Meier survival curves, which were plotted for each group. Each treated group was compared to the control using the two-sided log-rank test. In these analyses, *p* < 0.05 was regarded as significant. As these were exploratory analyses, we did not adjust *p* values for multiple comparisons.

## 3. Results

### 3.1. LR-IFN-β Gavage Lowers the Levels of Biomarkers of TBI-Induced Intestinal Damage

The effect of radiation on the expression of each of the 33 proteins representing the stress-response gene products and inflammatory cytokines was evaluated in the small intestine (ileum) and plasma (Table 1). These proteins have previously been shown to be useful biomarkers in the successful application of radiation mitigators [23,24]. In order to assess the effect of LR-IFN-β compared to *L. reuteri* regarding irradiation, we measured the radiation-induced changes in the intestine and plasma proteins. There was a radiation-induced decrease in the levels of several proteins in the intestines and plasma after WAI. The results on day 5 after irradiation, following a single fraction of 12 Gy TBI, are shown for three representative inflammatory cytokine proteins IFN-γ (Figure 2A), IL-3 (Figure 2B), and IL-17 (Figure 2C). The intestine levels of proteins showed changes induced by LR-IFN-β. These findings establish the effect that TBI has on the expression of cytokines, both in the intestine and systemically in plasma, after a single fraction of 12 Gy TBI.

TBI increased the intestinal level of INF-γ, which was significantly reduced by day 5 (postirradiation) by intraoral LR-IFN-β (*p* = 0.026) (Figure 2). The same trend was also noted when IFN-β was delivered through IP injections; however, the decrease occurred to a lesser extent and was not statistically significant relative to the level seen in the TBI mice. The radiation-induced decrease in IL-3 (Figure 2B) and increase in intestinal levels of IL-17 (Figure 2C) were significantly reversed by the intraoral gavage of LR-IFN-β (*p* < 0.007). Similarly, the effect was seen when the cytokine was delivered through an IP injection instead (*p* < 0.007). The irradiation caused a reduction in the intestinal levels of IL-3 (Figure 2B), a decline that was further exacerbated by the addition of IFN-β through IP injection as well as through the gavage of LR-IFN-β (*p* < 0.002). We also observed a 12,000-fold increase in the plasma level of G-CSF induced by irradiation (Appendix A) by day 5, following TBI (*p* < 0.001), which was significantly reduced following LR-IFN-β gavage. A similar trend was noted after the IP injection of IFN-β or gavage of *L. reuteri*; however, no statistical significance was reached compared to irradiation alone (Appendix A).

### 3.2. LR-IFN-β Gavage Reduces the Biomarkers of Intestinal Irradiation Damage

Occludins and I-CAM are intercellular adhesion proteins that act as biomarkers of an intact barrier that normally prevents bacterial entry into circulation. A single fraction of 12 Gy of TBI clearly decreased the levels of I-CAM (Figure 3) and occludin (Figure 4). The decrease was restored in those mice gavaged with LR-IFN-β 24 h postirradiation. The results between the mice that received the empty vector *L. reuteri* or the IP injections of IFN-β protein indicate that *L. reuteri* by itself and with soluble IFN-β produced some therapeutic effects, but the restorative effect was less than that observed in mice treated with gavage of LR-IFN-β. Next, we evaluated the mice that received 12 Gy TBI with respect to intestinal barrier function. We measured the effect of radiation and LR-IFN-β on the biomarkers of an intact intestinal barrier. Irradiation decreased the levels of both I-CAM and occludin. The gavage of LR-IFN-β ameliorated the radiation-induced decrease in I-CAM (Figure 3) and occludin (Figure 4). These results confirm a prior publication on the effect of gavage of LR-IL-22 on preserving I-CAM and occludin levels in irradiated intestines [18].

### 3.3. LR-IFN-β Ameliorates the Irradiation-Induced Decrease in Lgr5+ Green Fluorescent Protein (Lgr5+GFP+) in the Intestine of Mice following TBI and Stimulates the Biomarkers of Crypt Regeneration

The above results established that LR-IFN-β prevents the breakdown of intestinal barrier function after a single fraction of TBI. We next determined whether LR-IFN-β preserved the intestinal crypt stem cell numbers as another indication of radiation protection and mitigation during the TBI of Lgr5+ GFP+ mice (Figure 5) and crypt regeneration (Figure 6). LR-IFN-β ameliorated the radiation-induced loss of Lgr5+ GFP+ stem cells (Figure 5). There was an increase in crypt regeneration from LR-IFN-β treatment in the irradiated mice (Figure 6). Mice were injected 2 h before sacrifice with BrdU, a thymidine analog that acts as a marker of DNA synthesis and, thus, identifies proliferating cells. The sections were then stained with antibodies to BrdU. The levels of BrdU-positive crypts in the 0 Gy control group indicate the baseline turnover of the crypt cells during normal food consumption and house diet. The mice that received LR also showed increased numbers of BrdU-positive crypts (Figure 6) compared to the radiation group, but this was not significantly above the baseline 0 Gy levels.

### 3.4. LR-IFN-β Improves Survival of Mice after TBI or WAI

We evaluated the effects of the gavage of LR-IFN-β on the survival of C57BL/6 female adult mice subjected to one of two irradiation techniques: 9.25 Gy TBI or 19.75 Gy WAI. For the delivery of WAI to the mice, the thoracic cavity, head, neck region, and hind limbs were shielded. At 24 h post-irradiation, the mice were gavaged with either 200 µL saline or 200 µL saline containing 10^9^ bacteria LR-IFN-β in 200 µL of regular saline or control LR in 200 µL saline. With both TBI and WAI, the mice treated with LR-IFN-β delivered 24 h post irradiation had a significantly improved 30-day survival compared to the control irradiated mice in the two separate experiments (Figure 7 and Figure 8). No significant improvement in survival was detected in the mice undergoing oral gavage with control *L. reuteri* bacteria or IFN-β protein delivered IP 24 h after TBI (Figure 8). In other studies, IFN-β protein delivered 48 h after radiation produced improved survival [21]. The results establish that mice receiving gavage via LR-IFN-β had improved survival following radiation compared to the control irradiated mice (whether TBI or WAI). Figure 8 demonstrates that LR-IFN-β is more mitigative than either the IP injection of IFN-β protein or LR bacteria, without the IFN-β gene, following TBI.

### 3.5. Optimal Bacterial Dose Required for Gavage of LR-IFN-β

Next, we compared the bacterial cell numbers required for gavage to release IFN-β. Since the *L.* strain used *L. reuteri* (VPL1014) [25,26], which is a human gut symbiont, we did not expect to detect the bacterial colonization of the intestine, although the intestinal environment triggers bacteriophage-mediated lysis of engineered L. reuteri to release recombinant proteins [22,26]. The results showed that 10^9^ bacteria were required for the effective mitigation of intestinal radiation damage (Figure 9). LR-IFN-β numbers of 10^6^ up to 10^11^ bacteria in both male and female mice were compared to determine the optimum dose level for survival. The number of 10^9^ bacteria in 200 µL saline 24 h after 13.25 Gy PBI in the male mice and 13.4 Gy PBI in the female mice showed improved survival with better survival in females (Figure 9). These two figures demonstrate that the doses of bacteria we used in these experiments are the optimized doses of bacteria for the mitigation of WAI.

### 3.6. LR-IFN-β Delivered by Gavage Is Rapidly Cleared from the Intestine, and Bacterial Growth Is Not Detected in Plasma

Next, we evaluated LR-IFN-β clearance from the intestines and fecal matter. Following a single fraction of 13.354 Gy of PBI for female mice, the mice received the intraoral gavage of LR-IFN-β (Figure 10, Figure 11, Figure 12 and Figure 13). Groups of 5 mice were then sacrificed at days 0, 1, 2, 3, 4, or 5 after irradiation, and their intestinal tissue and fecal matter were evaluated for LR-IFN-β levels. Mice receiving the genetically engineered probiotic had significantly elevated levels of LR-IFN-β protein in their fecal matter and in the intestinal tissue at day 1, as determined by ELISA for IFN-β (Figure 10). No significant increases in IFN-β were found on the other days. No IFN-β protein was found in the other tissues including blood plasma, liver, kidney, or spleen. A colony-forming assay was used to detect the presence of LR-IFN-β in the fecal material, intestine, blood, liver, spleen, and kidneys. Colonies were detected in the fecal material and intestines on days 1 and 2 following the gavage of LR-IFN-β (Figure 11). No colonies were found in the blood taken at the same time, indicating no bacteria were detected in the blood or other tissues (Figure 12). In order to assess whether blood plasma inhibited the growth of LR-IFN-β, we plated LR-IFN-β on agarose gel alone or spiked with blood plasma. The colonies were noted to grow on all the plates, indicating that blood plasma does not impede the growth of LR-IFN-β in nonirradiated mice in Figure 13. Similar experiments were conducted with the spleen, kidneys, and liver tissue, and no bacterial colony growth was detected in any of the sampled materials. These data establish that LR-IFN-β is rapidly cleared from the intestine after gavage.

## 4. Discussion

The present results establish the effectiveness of the genetically engineered probiotic *L. reuteri*, which produces IFN-β (LR-IFN-β), as an intestinal protector and mitigator that will facilitate the delivery of therapeutic doses of whole abdominal irradiations. LR-IFN-β was administered by intraoral gavage and enabled transit through the stomach without being released into the blood prior to reaching the intestine. These data confirm prior work that demonstrated the effective use of antibiotic-resistant *L. reuteri* and plasmids to ensure the intestinal clearance of second-generation probiotics [22,25,26,29]. Once gavaged into the intestine, the IFN-β was released from the genetically engineered second-generation probiotic by bacterial lysis. The production of IFN-β is typically controlled by the cGAS-STING pathway, as described in [21]. The cGAS-STING signaling pathway is a cytosolic DNA-sensing pathway that is critical for host defense, which acts as a key mediator of inflammation in the settings of infections, cellular stress, and tissue damage following its activation by double-stranded DNA breakages [22]. The activated cGAS-STING pathway leads to the increased proliferation and inflammation of the intestinal stem cell niche (21).

The logic of using LR-IFN-β as a therapeutic in GI-ARS relates to its potential to ameliorate radiation toxicity. The rapid activation of the p53 pathway following radiation induced DNA damage, which suppresses delayed mitotic cell death [30,31]. This selectively activates cGAS-STING-dependent IFN-β production in the intestinal stem cell niche, which promotes intestinal stem cell proliferation and recovery following TBI [32]. Intestinal stem cell preservation and active regeneration through the delivery of IFN-β protein [21] were confirmed and extended by our present study in which we delivered LR-IFN-β to the Lgr5+ intestinal crypt stem cells. The IP injection of IFN-β protein or the intraoral gavage of LR-IFN-β revealed that the latter administration method was more effective. Our study closely simulates the clinical settings in which the WAI target volume includes the GI tract. The oral administration of a genetically engineered probiotic effectively released the cytokine after reaching the intestine and concentrated it locally at the intestinal crypts where the vulnerable Lgr5+ stem cells reside. In contrast, the gavage of the control *L. reuteri* did not provide protection.

The intravenous administration may not efficiently deliver IFN-β to the intestines following the radiation exposure. The success of LR-IFN-β as a therapeutic agent would be attributable to the local cytokine delivery to the intestinal crypts. Mechanistically, vascular swelling after each radiation fraction may have limited the parenterally administered drug from reaching the intestinal villi. Moreover, exogenous LR was shown to be rapidly cleared from the intestines and fecal matter following intraoral administration, with no evidence of colonization. A potential advantage of LR-IFN-β is the feasibility of developing a lyophilized LR-IFN-β formulation with high viability with the recombinant protein. This is already loaded inside the cell for the subsequent in vivo delivery. This would facilitate the possibility of LR-based drug administration.

We previously reported that LR-IL-22 gavage maintained the intestinal barrier function and modulated the release of irradiation-induced biomarkers in inflammation [18]. We demonstrated that the mechanism by which LR-IL-22 preserved the intestinal barrier function was by ameliorating the irradiation-induced decrease in the barrier proteins I-CAM and occludin in the ileum. In the present studies, LR-IFN-β administration reversed the elevated levels of several biomarkers of radiation injury, including IL-3, IL-17, and IFN-γ. After irradiation, the measured levels of the ICAM and Occludin proteins increased. Prior studies using LR-IL-22 revealed that the elevation in I-CAM and occludin correlated with reduced fluorescent bead leakage from the intestinal lumen into the blood [18]. The therapeutic effect of LR-IFN-β was also associated with the modulation of irradiation-induced levels of biomarkers in inflammation, including intestinal IL-3, IFN-γ, and IL-17. The intraoral gavage of the genetically engineered probiotic caused a significant reduction in all three intestinal cytokines. Interestingly, IL-3, a major cytokine produced by activated mast cells in the mucosal layer of the intestines [33], is significantly reduced in glucocorticoid-treated patients with inflammatory bowel disease (IBD) [33], emphasizing its importance in the immunomodulatory effect during inflammation. The fact that the control, irradiated mice had elevated levels of IL-3 that were reduced following the administration of LR-IFN-β highlights the importance of LR-IFN-β in modulating the immune system and maintaining intestinal barrier integrity. The cytokine Il-17 exerts strong proinflammatory activities, and its expression is increased in IBD patients [34]. Following irradiation, the intestinal expression of IL-17 was reduced by LR-IFN-β or IFN-β proteins. The IFN-γ-elevated levels after irradiation were significantly reduced in the irradiated mice following LR-IFN-β gavage. IFN-γ has been shown to drive disease pathogenesis in IBD patients by disrupting VE-cadherin-directed vascular barrier integrity [35].

These present data support a rationale for adding WAI to the regimens of chemotherapy and targeted therapies, including immune checkpoint inhibitors and the protocols for the treatment of advanced or recurrent epithelial ovarian cancer. The problem with WAI had been the inability to deliver a therapeutic dose due to intestinal toxicity. Combining immunotherapy with WAI should have a synergistic effect in ovarian cancer management, as the application of radiotherapy induces the in situ vaccination of tumor cells and apoptosis of Treg lymphocytes, which further enhances cell-mediated immune responses and cytotoxic T-cell activity [36,37,38,39,40,41,42,43,44,45]. Several clinical studies have also highlighted radiotherapy as a potential addition to the protocols for ovarian cancer. With intensity-modulated radiation therapy (IMRT) and stereotactic body radiation therapy (SBRT), there was hope for reduced toxicity; however, intestinal toxicity remained dose-limiting [6,7,8,9]. The present studies report a new method for reducing the toxicity of WAI in radiotherapy.

## 5. Conclusions

The present data support a rationale for translating the intraoral administration of LR-IFN-β to clinical protocols for use as mitigators of intestinal toxicity. If protection is achieved, the administration of LR-IFN-β may potentially allow women who are suffering from advanced, recurrent, or cisplatin-resistant disease to safely receive WAI. The addition of a safe, genetically engineered probiotic to a combined treatment modality using fractionated WAI and systemic chemotherapy and immunotherapy [46,47,48,49] regimens should facilitate a safe and effective protocol for reducing tumor burden, increasing survival, and improving the quality of life of patients with widespread abdominal ovarian cancer.

## Figures and Tables

**Figure 1 cancers-15-01670-f001:**
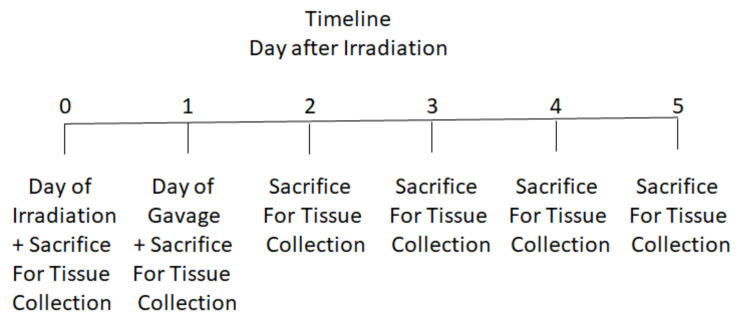
Timeline for experimental procedures.

**Figure 2 cancers-15-01670-f002:**
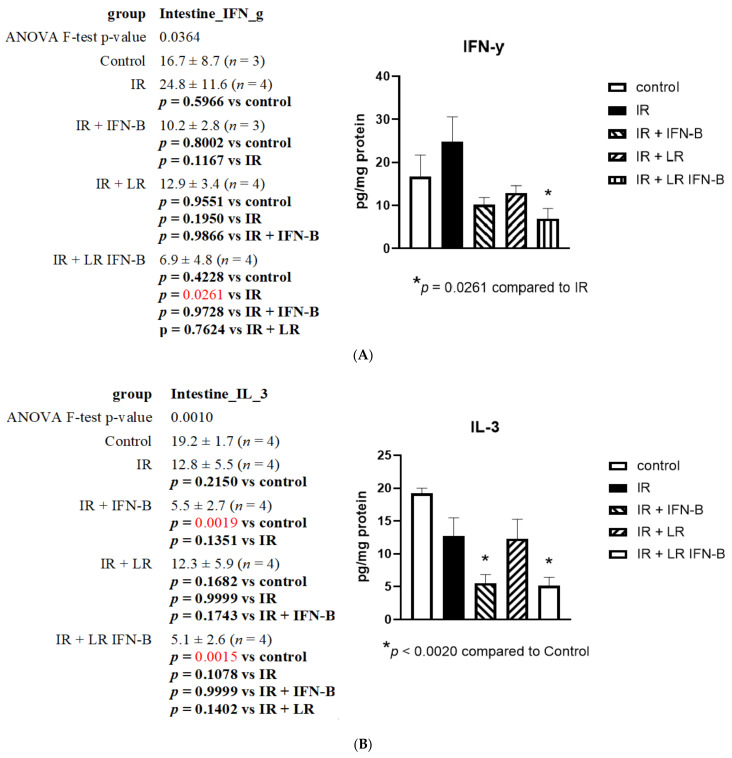
LR-IFN-β gavage ameliorates irradiation-induced changes in cytokine levels in the intestines: (**A**) IFN-γ; (**B**) IL-3; (**C**) IL-17. Five groups of mice (*n* = 3–4): control, irradiated (IR), IR plus intraperitoneal injection of IFN-β, IR plus intraoral 10^9^ bacteria copies of L. reuteri (LR) in 200 µL of saline, and LR, with copies of LR-IFN-β in 200 µL of saline. Mice received their respective treatment 24 h after irradiation to 12 Gy TBI. Intestinal tissues were analyzed on day 5 after TBI. Values are mean ± SEM. Total body irradiation (TBI). Red type indicates a significant difference.

**Figure 3 cancers-15-01670-f003:**
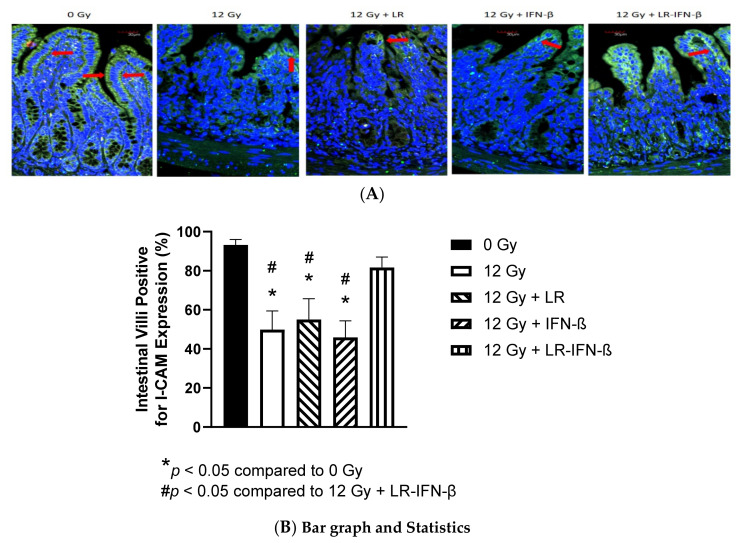
Reversal of the irradiation-lowered I-CAM in the intestinal crypts in LR-IFN-β-gavaged irradiated mice. C57BL/6 Lgr5+ mice (*n* = 25) were divided into 5 groups of 5 mice. The different groups were control, 12 Gy TBI, and 12 Gy TBI + LR, 12 Gy + IFN-β protein as an IP injection, and 12 Gy + LR-IFN-β. Mice in the third group received 10^9^ bacteria in 100 µL saline LR-IFN-β intraorally (gavage). Mice were then sacrificed 24 h after irradiation on day 5 for intestinal crypt cell analysis. (**A**) Immunostaining, as described in the Materials and Methods section, for levels in the intestine on day 5 after 12 Gy TBI. Green color is ICAM, and blue is DAPI staining identifying the nuclei. The red arrow is pointing towards the ICAM expressing cells (×200). (**B**) Quantitation. Total body irradiation (TBI). The * shows a significant *p* value comparing the treated groups with the 0 Gy group. The # indicates a significant p value comparing the other groups to 12 Gu + LR-INF-β.

**Figure 4 cancers-15-01670-f004:**
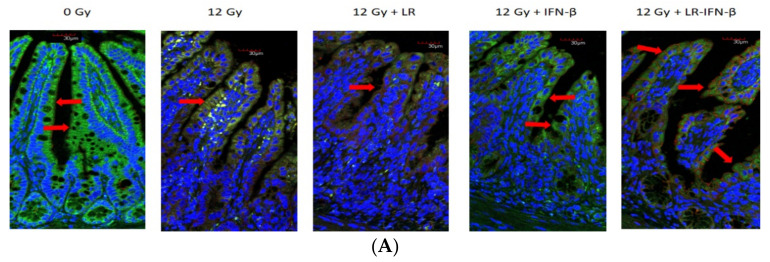
Reversal of the irradiation-lowered levels of occludin in the intestine of LR-IFN-β-gavaged irradiated mice. Increased occludin in the intestine of LR-IFN-β-gavaged irradiated mice. (**A**) Sagittal sections of intestine stained for occludin for each group (*n* = 5). Green stain is for occluding, while blue stain Dapi stain for nuclear DNA. Red arrow is point to the cells expressing occludins. (**B**) Quantitation of occluding-positive villi in 100 cross-sections of ileum from each group (×200) (*n* = 5). (12 Gy TBI). The * represents a significant difference between the treatment group and 0 Gy. The # represents a significant difference between the treatment group and 12 Gy + LR-IFN-β.

**Figure 5 cancers-15-01670-f005:**
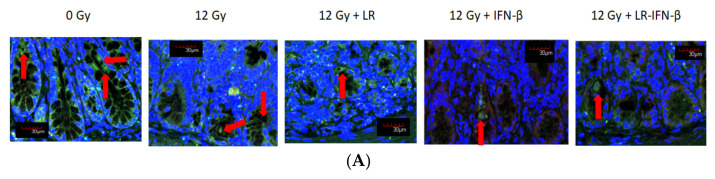
Preserved Lgr5+ GFP+ stem cells in the intestine of LR-IFN-β-gavaged irradiated mice. Increased Lgr5+ GFP+ stem cells in the intestine of LR-IFN-β irradiated (12 Gy TBI) mice. (**A**) Cross-sections of ileum from each group (×200). (**B**) Quantitation of the percentage of positive cells out of 100 cross-sections counted for each group of *n* = 5 for villi position. (**C**) Quantitation of number of crypts/sections scored. (**D**) Quantitation of number of GFP+ cells containing crypts scored. The * represents significant difference between the treatment groups and 0 Gy. The # represents significant difference between the treatment groups and the 12 Gy + LR-IFN-β group.

**Figure 6 cancers-15-01670-f006:**
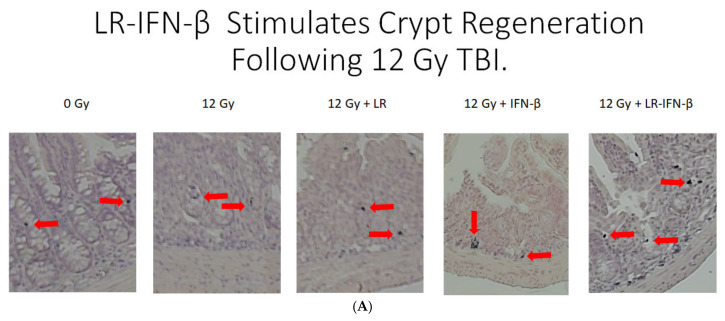
Increased BrdU-positive crypts in the intestine of LR-IFN-β-gavaged irradiated mice. Stimulation of crypt regeneration by LR-IFN-β following 12 Gy TBI. C57BL/6NTac female mice were irradiated to 12 Gy TBI, as previously described (18), and gavaged 24 h later with 10^9^ bacteria in 200 µL saline LR or LR-IFN-β or injected intraperitoneally with IFN-β (25 µg/kg in 100 µL saline). After 4 or 5 days postirradiation, the mice were sacrificed, and the intestines were removed, fixed in 10% formalin, and sectioned. Two hours before sacrifice, the mice were injected with BrdU (100 mg/kg) intraperitoneally. (**A**) The sections were stained with an anti-BrdU antibody. The red arrows are pointing to BrdU positive cells. (**B**) The number of regenerating crypts in the 100 cross-sections/ileum for each group (*n* = 5) of mice were counted. Mice gavaged with LR-IFN-β had an increased number of regenerating crypts by increased staining for BrdU. The * represents significant difference between the treatment groups and 0 Gy. The # represents significant difference between the treatment groups and the 12 Gy + LR-IFN-β group.

**Figure 7 cancers-15-01670-f007:**
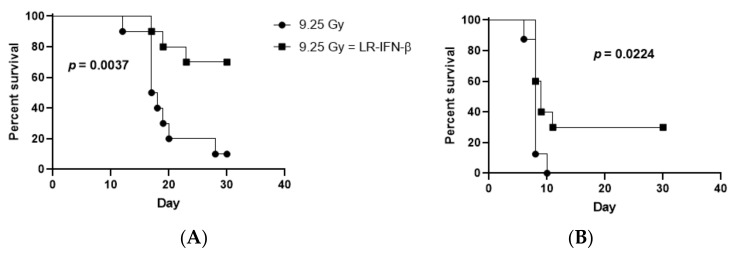
LR-IFN-β gavage increases the survival of both TBI- (**A**) and WAI-irradiated mice (**B**). C57BL/6 mice received either 9.75 Gy TBI (**A**) or 19.75 Gy WAI (**B**) and were then gavaged 24 h later with 200 µL of regular saline or 200 µL of 10^9^ bacteria LR-IFN-β. Mice were followed for 30 days, and survival was monitored among the different treatment modalities. (*n* = 10). Total body irradiation (TBI). Whole abdomen irradiation (WAI).

**Figure 8 cancers-15-01670-f008:**
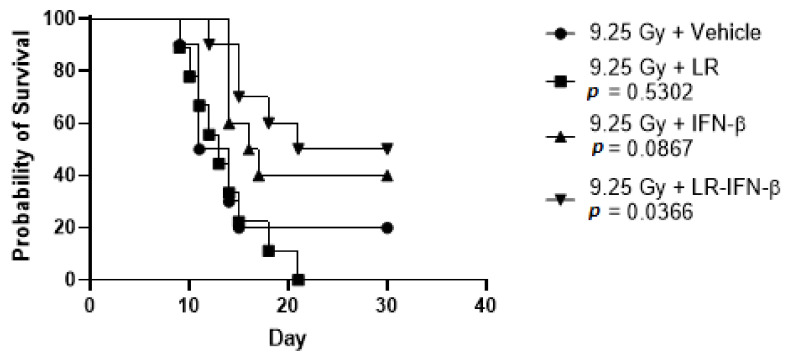
Increased survival of 9.25 Gy TBI mice by LR-IFN-β gavage 24 h after irradiation (*n* = 10 mice per group).

**Figure 9 cancers-15-01670-f009:**
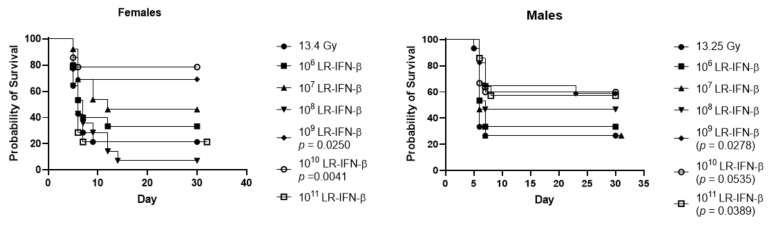
Determination of the optimal number of gavaged LR-IFN-β bacteria required to increase survival in female and male mice after 13.4 or 13.25 Gy PBI, respectively.

**Figure 10 cancers-15-01670-f010:**
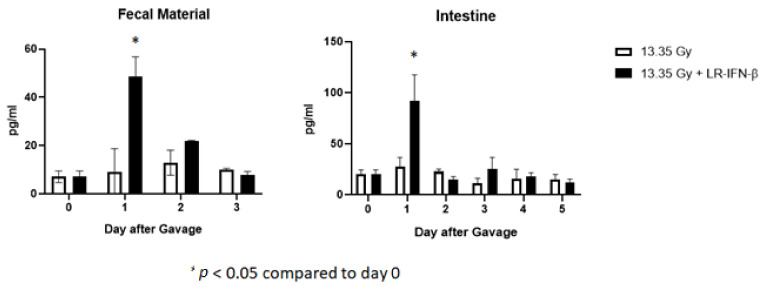
Increased IFN-β in fecal material and intestine from female mice irradiated to 13.35 Gy PBI following the gavage of LR-IFN-β, as determined by IFN-β ELISA. Increased concentration of IFN-β was detected in the fecal material and intestines one day after the gavage of LR-IFN-β. No increased IFN-β was detected in the blood or other tissues. Values are mean ± SEM. demonstrates a significant difference when compared to day 0.

**Figure 11 cancers-15-01670-f011:**
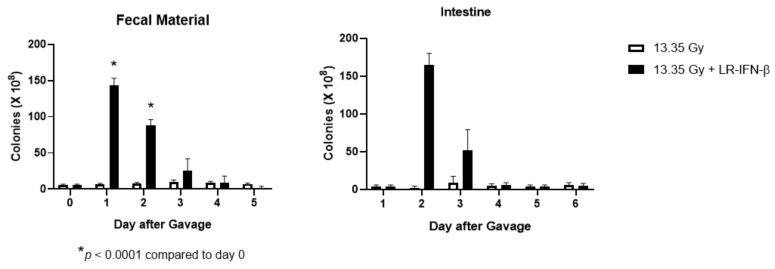
Clearance of LR-IFN-β from female mice feces and intestine after 13.35 Gy PBI and gavage of 10^9^ LR-IFN-β bacteria: (levels at day 0 through to 5). * Demonstrates a significant difference when compared to day 0.

**Figure 12 cancers-15-01670-f012:**
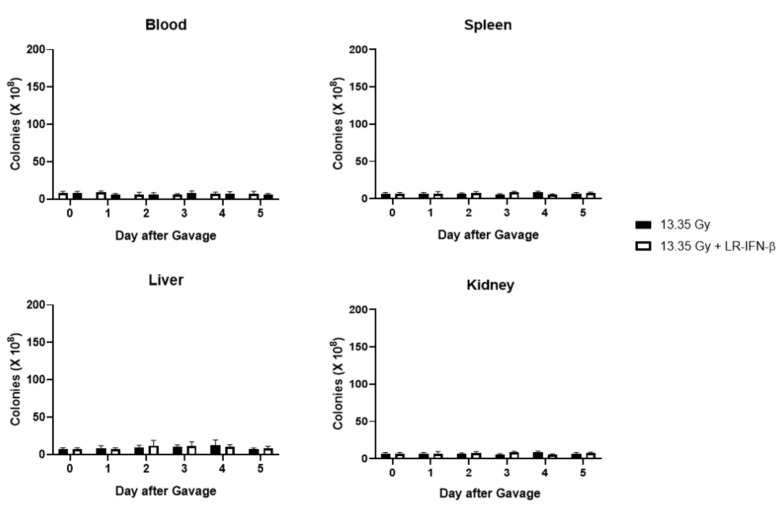
LR-IFN-β was not detected in blood, spleen, liver or kidney following radiation and gavage of LR-IFN-β. Female mice were irradiated to 13.35 Gy PBI and gavaged with LR-IFN-β 24 h after irradiation. Mice were sacrificed on days 0, 1, 2, 3, 4 and 5 after gavage. Blood, spleen, liver, and kidney were removed and assayed for LR-IFN-β. No bacteria were detected.

**Figure 13 cancers-15-01670-f013:**
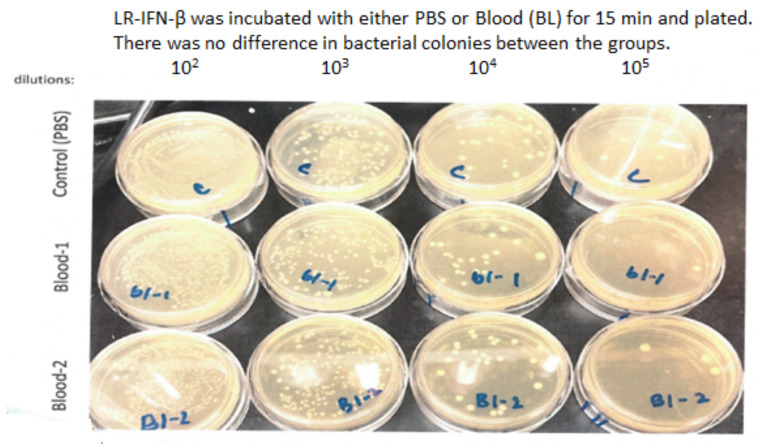
No detectable LR-IFN-β in the blood of gavaged and irradiated mice. To determine if blood plasma inhibited the growth of LR-IFN-β bacteria on agarose plates, LR-IFN-β was incubated with 100 µL PBS or with 100 µL blood plasma on agarose plates containing 20 µg/mL erythromycin for 15 min. Colonies grew on all the plates, and there was no difference in bacterial colonies between groups.

**Table 1 cancers-15-01670-t001:** Protein expression analyzed by Luminex assay.

Protein	Function
TGF-β	Inflammation, pro-fibrotic controls proliferation, and cellular differentiation
TNF-α	Inflammatory cytokine-stimulates necroptosis
IL-1α	Pro-inflammatory
IL-1β	Pro-inflammatory
IL-2	Natural response to microbial infection: regulates activities of white blood cells
IL-3	Stimulates hematopoietic stem cells to become myeloid progenitor cells
IL-4	Stimulation of activated B-cell and T-cell proliferation, and stimulates B cells into plasma cells
IL-5	Stimulates B cell growth and increases immunoglobulin secretion
IL-6	Pro-inflammatory
IL-7	Stimulates hematopoietic stem cell differentiation into lymphoid progenitor cells
IL-9	Stimulates cell proliferation and prevents apoptosis
IL-10	Anti-inflammatory
IL-12 (p40)	Stimulates growth and function of T cells and stimulates IFN-γ and TNF-α
IL-12 (p70)	Stimulates growth and function of T cells and stimulates IFN-γ and TNF-α
IL-13	Mediator of allergic inflammation and induces MMPs
IL-15	Induces proliferation of natural killer cells
IL-17	Recruits monocytes and neutrophils to sites of inflammation
IP-10	Chemoattractant for monocytes/macrophages, T cells, NK cells, and dendritic cells
KC	Attracts neutrophils
LIF	Stem cell differentiation
LIX	Cell migration and activation of neutrophils
MCP-1	Recruits monocytes, T cells, and dendritic cells to sites of inflammation
M-CSF	Induces hematopoietic stem cells to differentiate into macrophages
MIG	T-cell chemoattractant
MIP-1α	Recruitment and activation of granulocytes
MIP-1β	Chemoattractant for NK cells, monocytes, and other immune cells
MIP-2	Recruits neutrophils and lymphocytes in the intestine
RANTES	Recruits leukocytes into inflammatory sites
VEGF	Stimulates vasculogenesis and angiogenesis
Eotaxin	Recruits eosinophils
GM-CSF	Stimulates hematopoietic stem cells to produce granulocytes (neutrophils, eosinophils, and basophils), and monocytes
G-CSF	Stimulates hematopoietic stem cells to produce granulocytes
IFN-γ	Activator of macrophages and inducer of class II major histocompatibility complex (MHC) molecule expression

## Data Availability

Data is available in Supplementary Data.

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
