# Peer review of "Release of Interferon-β (IFN-β) from Probiotic Limosilactobacillus reuteri-IFN-β (LR-IFN-β) Mitigates Gastrointestinal Acute Radiation Syndrome (GI-ARS) following Whole Abdominal Irradiation"

_cancers, 2023, doi:10.3390/cancers15061670_

Round 1
Reviewer 1 Report
Hamadea et al in their study demonstrate the significance of Probiotic based management of radiation syndrome which is important area and need intense investigation in future for developing the area.
I think the study is competent and bear merits however I have following concerns which need to be addressed by the authors
1. The authors start the manuscript with ovarian cancer but did not work with OA tumor cells. So I would advice to remove the OC and related text in the manuscript. They should only focus on gastric syndrome
2. title need to be changed completely and it should only focus on managing radiation syndrome
3. Now to the experiments , It is clear that LR-driven IFN beta is normalizing lethally irradiated intestine better then purified IFN-beta but from their experiments it is not clear why ? This is a lacuna in the study.
4. Although author have provided the panel of important cytokines which are important for the intestinal damage and fibrosis like IFN gamma , IL-17 but they did not provide the status of Th2 cytokines which are important for tissue reconstitutions. So I would suggest author to add ratio of Th1/ Th2 panel which would be more conclusive to corelate the status of gut immune homeostasis.
5. It would be advantageous to corelate their findings with the activation and deactivation of CD44+ paneth cells and CD169+/ TCR-1/ CD206+ intestinal macrophages and M cells.
6. Their study showed the influence of LR-IFN beta on the stem cells which is fine but what is actually stimulating these cells is not discussed in their experiments
7. What figure 7 and 8 are adding to is not clear.
8. Whether author tried doing reconstitution analysis to address the underlying mechanism. would be good to know
9. Can author check the impact on LR-IFN beta on the activation of peritoneal macrophages for correlation with immune polarization paradigm
Author Response
See attached response letter.

Reviewer 2 Report
The manuscript titled "Intraoral Gavage of Genetically Engineered Probiotic Limosilactobacillus reuteri Releasing IFN-β Mitigates Intestinal Irradiation Toxicity" is an original work done by the authors as a continuum to their previous work. Although the work presents interesting findings. The Authors need to address several major issues:
1. The authors used an engineered bacteria LR-IFNb to deliver the cytokine specifically to the intestinal crypts. The authors have briefly described the production of the engineered strain but failed to show its functionality and interferon production dynamics. For example, the authors haven't presented any data on IFNb production by the bacteria nor did they showed what dose of IFNb this bacterial strain was able to deliver to the mouse intestine. The inconsistent outcome of the bacterial dose escalation study in female mice receiving 13.35Gy radiation might specifically point to this issue.
2. An experimental time line showing the procedures and evaluations at each time point will make it easy to understand the study.
3. Table 1 can be moved to supplementary and additing references to the table will be useful for readers.
4. Figure 1 is showing data on TBI, but WAI being mentioned in the results addressing Figure 1 and in Figure 1 caption is confusing.
5. Along with figure 6 and 7 please provide body weight changes of the animals and specifically mention the reason for the death or euthanesia of the mice. Specifying the reason for the death of the mice will increase credibility of the results.
6. The authors claim, the LR-IFNb produces IFNb - the data for this is missing, given the fact that an expected effect was observed (Secondary evidence only). However this point need to be verified with primary evidence.
7. The authors claim "Oral administration of a genetically engineered probiotic effectively released the cytokine after reaching the intestine and concentrates it locally at the intestinal crypts". This authors haven't provided this data hence this cannot be claimed.
Author Response
See attached response letter.

Round 2
Reviewer 1 Report
Although Authors have addressed my comments but I would like to see another story from the author addressing the comments which can not addressed at the moments
Reviewer 2 Report
I appreciate the authors for addressing the comments and improving the manuscript.